# Phylogeny and Virulence Factors of *Escherichia coli* Isolated from Dogs with Pyometra

**DOI:** 10.3390/vetsci9040158

**Published:** 2022-03-25

**Authors:** Roberta T. Melo, Raquel P. Oliveira, Beatryz F. Silva, Guilherme P. Monteiro, João Paulo E. Saut, Letícia R. M. Costa, Sthéfany Da C. Dias, Daise A. Rossi

**Affiliations:** 1Laboratório de Biotecnologia Aplicada, Faculdade de Medicina Veterinária, Universidade Federal de Uberlândia, Uberlândia 38402-018, MG, Brazil; guil.paz@hotmail.com (G.P.M.); leticia.m.costa@unesp.br (L.R.M.C.); sthefany.c.dias@unesp.br (S.D.C.D.); daise.rossi@ufu.br (D.A.R.); 2Hospital Veterinário, Faculdade de Medicina Veterinária, Universidade Federal de Uberlândia, Uberlândia 38402-018, MG, Brazil; raquelperes_o@yahoo.com.br (R.P.O.); betrayzfonseca@uft.edu.br (B.F.S.); 3Laboratório de Saúde em Grandes Animais, Faculdade de Medicina Veterinária, Universidade Federal de Uberlândia, Uberlândia 38402-018, MG, Brazil; jpsaut@famev.ufu.br

**Keywords:** *E. coli*, canine pyometra, urine, uterus, virulence genes

## Abstract

We aimed to investigate the occurrence, phylogeny, and virulence of *E. coli* in the uterine contents and urine of female dogs with pyometra, through the presence of virulence genes and their genetic similarity. Uterine secretions and urine samples from 52 female dogs with pyometra were collected and cultured. Strains identified as *E. coli* from 25 uterine and 7 urine samples were tested for virulence genes by PCR. Genetic similarity between the isolates was studied using RAPD-PCR. *E. coli* was observed in 48.07% uterine samples with pyometra and 20.0% urine samples. The strains showed high percentages for the presence of virulence genes: 96.9% had the gene *sfa*, 59.4% *afa,* 46.9% *pap*, 53.1% *hly*, and 68.75% *cnf*. Even with the high prevalence of virulence genes, the samples were not submitted to DNA sequencing to confirm the results. Analysis showed high genetic diversity in *E. coli,* however, strains isolated from the same animal indicate that cystitis and pyometra could be related. Our study indicated the association between *E. coli* in dogs with pyometra and cases of urinary tract infection and the pathogenic potential of strains increasing with animal age.

## 1. Introduction

Canine pyometra, also known as a cystic endometrial hyperplasia-pyometra complex, is a disease characterised in the adult dog by uterine inflammation and exudate accumulation. This disease is caused mainly by opportunistic bacteria, and high levels of hormones in the body, such as progesterone and oestrogen, may be preconditioning factors [1,2,3,4]. The incidence of pyometra in female dogs is high, and this disease can be potentially fatal [5].

The most commonly isolated bacteria from the uterine secretions of female dogs with pyometra are *Escherichia coli* [6,7]. The genetic similarity between *E. coli* strains isolated from female dogs with simultaneous urinary tract infection (UTIs) and pyometra, showing that the same bacterium infects both sites, has already been demonstrated [8].

*E. coli* isolates from dogs with pyometra and/or cystitis may have virulence characteristics that include the presence of the *pap* (P-fimbriae encoding), *afa* (adhesin afimbrial), *sfa* (S-fimbriae), *hly* (α-haemolysin), and *cnf* (cytotoxic necrotic factor) genes. The presence of these genes demonstrates the invasive potential of the strains [9,10,11,12,13].

Therefore, this study aimed to evaluate the occurrence of *E. coli* in both the uterine secretions and urine of female dogs with pyometra and to analyse these isolates regarding the presence of virulence factors and genetic similarity.

## 2. Material and Methods

Fifty-two uterine and 35 urine samples were collected from female dogs (*Canis familiaris*) with pyometra. Samples were collected in 2013 from female patients at the Veterinary Hospital of the Faculty of Veterinary Medicine, Federal University of Uberlândia (FAMEV-UFU), Uberlândia, Minas Gerais, Brazil. The project was approved by the University’s Ethics Committee on the Use of Animals with the number 046/13.

A diagnosis of pyometra was made based on physical examination, haemogram results, and ultrasound [14]. At the time of ovariohysterectomy surgery, the uterus was removed from the animal, kept in a cooler, and sent immediately to the Laboratory of Molecular Epidemiology of FAMEV–UFU [15]. The uterine secretion was aspirated with a sterile needle gauge 40 × 12 mm coupled to a sterile 20 mL syringe. Urine collection was performed using the cystocentesis technique [16]. Both uterine secretions and urine were subjected to bacterial culture to search for the presence of *E. coli*.

Microbiological analyses were performed [6], with the culture media Sheep Blood Agar 5% (Oxoid, SP, Brazil Ltd.a), MacConkey agar (Oxoid, SP, Brazil Ltd.a), and BHI (Brain and Heart Infusion Broth; Oxoid, SP, Brazil Ltd.a; incubation at 37 °C for 24 h under aerobic conditions). A commercial kit was used for the identification of *E. coli* by biochemical profiling (tryptophan production, glucose fermentation, gas production from glucose, hydrogen sulphide production, lysine use, ornithine use, motility, indole production, rhamnose, and citrate use) (Laborclin, Pinhais, Paraná, Brazil). *E. coli* strain ATCC 25922 (MicroBioLogics, St. Cloud, MN, USA) was used as a positive control in phenotypic tests.

Strains identified as *E. coli* were subjected to the identification of virulence genes. We first assessed DNA extraction by thermal lysis. For DNA extraction, 2 mL cultures were transferred to microtubes (Bio express, SP, Brazil) and centrifuged (Cientec^®^, Belo Horizonte, Minas Gerais, Brazil) at 12,000× *g* for 5 min to form a pellet. The supernatant was discarded and 200 µL of phosphate buffer plus 2.5 µL of the protease solution (DuPont™ PCR Reagent, Wilmington, Delaware, EUA) were added to the pellet. This mixture was heated at 37 °C for 20 min and at 95 °C for 10 min in a thermocycler (Eppendorf^®,^ SE, Germany). It was then transferred to a cooling block (2 °C to 8 °C) for 5 min to obtain the DNA. An aliquot of the supernatant was used as a DNA template in the PCR [17] and quantification in the Nanodrop device (Thermo Fisher Scientific, SP, Brazil) [18]. *Primer* pairs and references to the amplification protocol and positive controls are shown in Table 1. The negative control was composed of sterile ultrapure water, which was added to the reaction mixture instead of DNA.

For amplification of the *sfa*, *pap*, *hly*, and *cnf* genes, the final volume of the reaction (50 µL) contained 50 ng of the bacterial DNA solution and the following reagents: 10 mM of Tris-HCL; 50 mM KCl; 200 µM of each deoxynucleotide triphosphate (DNTP); 1.5 mM MgCl_2_; 20 picomoles of the *sfa*, *pap*, and *cnf* primers and 30 picomoles of the *hly* primers, along with 0.2 U of Taq DNA polymerase (Invitrogen^®^, São Paulo, Brazil). Each gene was studied separately in the reactions. Amplification was performed in a thermocycler (Eppendorf^®^, Hamburg, Germany), with the following cycles: an initial cycle at 95 °C for 5 min, 35 amplification cycles, consisting of denaturation at 95 °C for 1 min, annealing at 55 °C for 1 min, extension at 72 °C for 1 min, and a final cycle of extension at 72 °C for 5 min [17]. For the *afa* gene, the amplification solution consisted of 10 mM of Tris-HCL, 50 mM KCl, 2.0 mM MgCl_2_, 200 µM of each deoxynucleotide triphosphate (DNTP), 30 picomoles of primer, 1 U of Taq, and 70 ng of DNA in a final volume of 30 µL. Amplification was performed in a thermocycler (Eppendorf^®^, Hamburg, Germany) with the following cycles: an initial cycle at 94 °C for 10 min, 30 amplification cycles of denaturation at 94 °C for 1 min, annealing at 63 °C for 1 min, extension at 72 °C for 2 min, and a final cycle of extension at 72 °C for 7 min [22].

The genetic similarity between *E. coli* isolates was assessed using Random Amplification of Polymorphic DNA PCR (RAPD-PCR) with the *primers* 1247 (5′AAGAGCCCG3′) and 1290 (5′GTGGATGCGA3′) [10]. Gels containing the amplified products were taken to capture images (Loccus Biotechnology, SP, Brazil) and computational analysis was performed using the GelCompar II Program (https://www.applied-maths.com/gelcompar-ii—Comparative Analysis of Electrophoresis Patterns. Accessed on 8 November 2016). Bands of weak, medium, and strong intensity captured by the program were considered in the analysis. A similarity matrix was obtained by comparing pairs of strains using the Dice similarity coefficient, adopting 1% tolerance for each *primer* separately. The final analysis was based on the average of experiments. The unweighted pair group method with the arithmetic mean (UPGMA) method was used for the construction of dendrograms comprising all of the studied strains.

All results were tabulated and analysed for descriptive statistics, calculating the percentage of isolation and presence of virulence genes in isolates identified as *E. coli*.

## 3. Results

Bacterial isolation was achieved in 36/52 (69.2%) uterine secretion samples of studied dogs. Of the 35 urine samples collected from the pyometra dogs, only 10 (28.6%) showed bacterial growth.

Regarding *E. coli* positivity, it was observed that 25/52 (48.07%) uterine secretion samples and 7 of the 35 (20.0%) urine samples contained the bacteria, totalling 32 strains. It showed that animals with positive urine samples for *E. coli* also showed positive uterine secretions, except in one animal where the agent was isolated in the urine only. Thus, we characterised 32 strains of *E. coli* from 26 animals of 9 different breeds with ages varying from 1 to 14 years (Table 2).

The presence of the *sfa* gene was found in 31/32 *E. coli* isolates (96.9%) of which 24 were from uterine samples and 7 from urine. The results of the present study were also higher than other studies [23,24] about the *afa* gene, which was present in 19/32 of the isolates (59.4%), with 15 being from uterine samples and 4 from urine. The *pap* gene was detected in 15/32 strains (46.9%), of which 11 were derived from uterine secretion and 4 from urine. For the *hly* gene, positivity was found in 17/32 isolates (53.1%), of which 13 were from uterine secretion and 4 from urine. The last reported gene *cnf* was detected in 22/32 isolates (68.75%), of which 17 were from uterine secretions and 5 from urine. Of the total strains, 11/32 (34.4%) had all of the virulence genes (V10) investigated and therefore a greater pathogenic potential (Table 3).

Four distinct groups (1–4) were identified (Figure 1), with proximity between 51% and 64%, which indicates high variability within the groups. In addition, four genotypes can be observed with genetic similarity lower than 50%, corresponding to the isolates from the uterine secretions of animals 12, 14, 18, and 23. In Group 1, with 53.8% homology, we determined the presence of two *clusters* “a” and “b” with similarities of 88.9% and 93.8%, respectively. Each *cluster* consisted of two strains originating from the same animal (05 and 01, respectively) and distinct samples of the uterine secretions and urine. In Group 2, with 55.9% homology, two *clusters* “c” and “d” were detected with a genetic similarity of 90.0% and 88.9%, respectively. Both *clusters* also presented two strains isolated from the same animal (02 and 06, respectively). Compared to Group 3, with 56.4% homology, we identified only one *cluster* “e” with a genetic similarity of 87.3%, consisting of two strains from female dog number 04. Group 4, with the homology of 63.9%, was composed of two strains with a distinct pattern.

## 4. Discussion

The complex pathogenesis of pyometra is not fully understood, but it involves hormonal and bacterial factors. Regarding factors related to bacterial pathogens, we can include the species and characteristics of virulence [25].

Our study found *E. coli* positivity was observed in 25/52 (48.07%) uterine secretion samples and 7/35 (20.0%) urine samples contained the bacteria, totaling 32 strains. A high rate of isolation of this pathogen has been previously reported in female dogs with pyometra in 43/48 (90.0%) of samples [2]. On the prevalence of *E. coli*, researchers from Brazilian veterinary hospitals [14,21,26,27,28,29,30,31,32] and the United States [31] found percentages close to those found here, between 57.0% and 73.0% for *E. coli* isolation.

In relation to the virulence factors, the *sfa* gene, our findings were higher compared to a previous study [33]. These authors reported positivity for this gene in 22/51 (43.1%) of the *E. coli* isolates from UTIs in female dogs and 24/52 (46.1%) of the uterine secretion samples of animals with pyometra. Studies conducted in Brazil demonstrated the presence of this gene in 19/33 (56.0%) and 120/151 (79.5%) strains of *E. coli* isolated from female dogs with pyometra [6,32]. S-fimbriae, encoded by the *sfa* gene, is composed of several subunits that are extremely important in the interaction of the bacteria with the epithelial cells of the kidneys and bladder [34]. These fimbriae are often found in isolates involved in sepsis. Thus, with septicemia being one of the consequences of pyometra, the presence of this virulence factor may contribute to the dissemination and generalisation of infectious conditions [6].

The presence of the *afa* gene in our research was also higher than that reported by other authors [6], who observed this gene in only 5/151 (3.3%). *E. coli* isolates has found that the adhesin encoded by this gene plays a small role in uterine colonisation. This low frequency was also reported [27], with 1/52 (1.9%) of the pyometra and 1/51 (2.0%) of the UTI isolates presenting *afa.*

The *afa* gene is not often found in *E. coli* isolated from uterine secretions of female dogs with pyometra [35]. This gene is more associated with uropathogenic *E. coli* isolated from humans [36]. According to previous work, the high percentage of *afa* positivity is associated with contact between humans and dogs [37]. These authors observed a 95% genetic similarity in *E. coli* isolated from samples of canine faeces and their owners.

The *pap* gene is of great clinical importance because it is involved in the synthesis of pili P, which is the most important adhesin in those strains that cause renal infections [38]. The *pap* gene was found in 87/151 (57.6%) strains of *E. coli* isolated from female dogs with pyometra [6].

The synthesis of haemolgysin is regulated by the *hly* gene. This cytolysin is capable of damaging erythrocytes, leukocytes, and endothelial cells of the kidneys, which favours infection [35,36,39]. During erythrocyte lysis, there is the absorption of exogenous iron, an essential element for bacterial metabolism. This is possible through the synthesis of exoproteins, recognised as siderophores by microorganisms [38].

For the *hly* gene, positivity was found in 17/32 (53.1%) isolates. Lower results were found in Brazil [34], where it was observed in 17/51 (33.3%) and 18/52 (34.6%) positive isolates in UTI and pyometra from dogs, respectively. In other countries, this gene has been reported in 13/23 (52.0%) strains of *E. coli* derived from female dogs with pyometra in Australia and 7/30 (23.3%) strains of *E. coli* isolated from the urine of dogs in Italy [27,35].

The “cytotoxic necrotising factor” produced by the *cnf* gene was only associated with uropathogenicity due to epidemiological findings [27]. However, culture studies of human neutrophils have indicated that this toxin may influence the immune response of the host since it seems to allow bacterial death by neutrophils [30]. This gene encodes an important cytotoxin that facilitates the spread of bacteria from the lumen of the bladder and intestinal tract into the bloodstream [36]. Considering the importance of this gene in the invasive potential of *E. coli*, its presence can facilitate infections in different places, including uterine infections. Studies in Brazil and countries such as the United States and Italy found low percentages of *cnf* in *E. coli* isolated from the uterine secretions of female dogs with pyometra. The level of variation was 21.6–57.0% in Brazil and 41.0–53.3% elsewhere [6,32,33,40]. Our study shows high prevalence rates identified for the virulence genes in the samples evaluated and encourages the importance of confirmation using the genomic sequencing technique.

The analysis of genetic similarity presented in the dendrogram (Figure 1) of *E. coli* isolates demonstrates high genetic diversity, possibly due to the different environments, management, and age of the animals in the study. The genotypic diversity demonstrated suggests the possibility of genetic changes in different lineages and the probability of the emergence of a new genotype in the region [12,41]. Several factors enable the evolution of bacteria, such as horizontal transfer, which facilitates adaptation to new environments. This contributes to the acquisition of virulence factors directly involved in infections and the development of different clusters [7].

The relationship between urine isolates and uterine secretion from the same animal (Table 2) was shown by the genetic similarity of the *E. coli* strains (Figure 1). Most of the strains present in the same animal were of the same genotype (5/6 83.3%). Thus, it can be inferred that cystitis and pyometra in animals 01, 02, 04, 05, and 06 had a direct relationship and that these microorganisms probably showed tropism to both organs (the bladder and uterus).

Moreover, this pattern of high proximity between strains isolated from the same animal was also evidenced by the similarity of the genotypic characteristics between strains. For animal 4 (cluster “e”), it was observed that both strains were positive for all virulence genes tested (*sfa*, *afa*, *pap*, *hly*, and *cnf*). The *sfa*, *pap*, *hly*, and *cnf* genes were present in the bacteria of animal 02, and the *sfa* gene in the *E. coli* of animal 06. For animal 01 (cluster “b”) the strains were possessed in the *cnf*, *sfa*, and *afa* genes, while for dog 05, cluster (“a”), the *E. coli* isolates showed the *sfa* gene.

This genetic similarity observed between urinary bacteria and uterine secretion of the same animal had already been observed [31]. In a previous study, 14/16 (87.5%) *E. coli* strains isolated from UTIs have been reported to be identical or similar to those isolated from the infected uterus. They also identified similarities between uterine and faecal bacteria [42]. Interestingly, these authors concluded that *E. coli* associated with canine pyometra would originate from the faecal microbiota and that UTIs would [43] have occurred from the same *E. coli* clone observed in the uterus, considering that the isolates were similar but not identical. Any disequilibrium of microflora residing around the urethral ostium of the vagina, especially *Lactobacilli*, would be a good opportunity for the colonisation of the urinary tract by *E. coli* or other potentially pathogenic agents [38].

Future work should be performed to determine the actual zoonotic importance of *E. coli* isolated from female dogs by investigating their similarity to human isolates. The studies should determine whether they have common characteristics, such as resistance to antimicrobials and the presence of resistance genes, to verify whether there is a transfer of virulence to the *E. coli* strains of humans. Such research could bring greater insight and could help more comprehensible actions to prevent risks to humans and animals.

The monitoring of *E. coli* isolates from female dogs for virulence genes is important because the same strains that cause infections in dogs can also infect humans, thus indicating their importance for public health.

## 5. Conclusions

This study corroborated the importance of *E. coli* as one of the major aetiologic agents of canine pyometra. The genotypic diversity observed in *E. coli* isolated from different animals demonstrated that canine uterine and urinary infections arise from many, not interconnected sources. However, the genetic similarity between strains from the same animal indicates a relationship between the two types of infection. The presence of several virulence factors, and the high prevalence of these genes, demonstrate the pathogenic potential of these strains.

## Figures and Tables

**Figure 1 vetsci-09-00158-f001:**
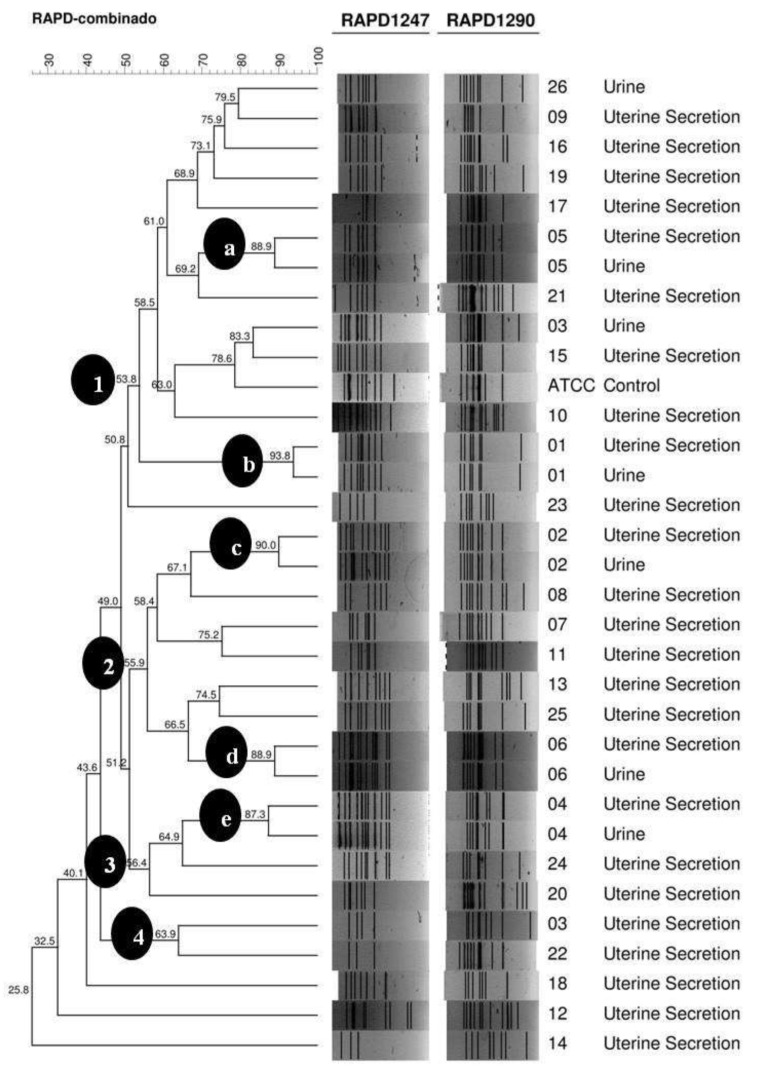
Dendrogram of 32 *E. coli* isolates originated from female dogs with pyometra Uberlândia—MG, by RAPD-PCR with primers 1247 and 1290, using the average of experiments with a tolerance of 1.5% and UPGMA method with optimisation of 85% at GelCompar II program, Sint-Martens-Latem, Belgium. Profile 1—a group with 53.8% homology, consisting of two clusters (a and b) with 88.9% and 93.8% of similarity, respectively. Profile 2—a group with 55.9% homology, consisting of two clusters (c and d) with 90.0% and 88.9% similarity, respectively. Profile 3—a group with 56.4% homology, composed of the cluster (e) with 87.3% similarity. Profile 4—a group with 63.9% homology.

**Table 1 vetsci-09-00158-t001:** *Primers* for identification of virulence genes *pap*, *hly*, *cnf*, *sfa*, and *afa* in *Escherichia coli*.

Genes	Sequence 5′→3′	Molecular Weight	Positive Control
*pap-1*(R)	GCAACAGCAACGCTGGTTGCATCAT	336 bp [19] ^a^	*E. coli* J96 [20,21] ^c^
*pap-2*(F)	AGAGAGAGCCACTCTTATACGGACA		
*hly-1*(R)	AACAAGGATAAGCACTGTTCTGGCT	1177 bp [19] ^a^	*E. coli* J96 [20,21] ^c^
*hly-2*(F)	ACCATATAAGCGGTCATTCCCGTCA		
*cnf-1*(R)	AAGATGGAGTTTCCTATGCAGGAG	498 bp [19] ^a^	*E. coli* J96 [20,21] ^c^
*cnf-2*(F)	CATTCAGAGTCCTGCCCTCATTATT		
*sfa-1*(R)	CTCCGGAGAACTGGGTGCATCTTAC	410 bp [19] ^a^	*E. coli* J96 [20,21] ^c^
*sfa-2*(F)	CGGAGGAGTAATTACAAACCTGGCA		
*afa-1*(R)	GCTGGGCAGCAAACTGATAACCTC	750 bp [20] ^b^	*E. coli* A30 [21] ^c^
*afa-2*(F)	CATCAAGCTGTTTGTTCGTCCGCCG		

(R): reverse. (F): forward. ^a^ [19]; ^b^ [20]; ^c^ [21] strains kindly provided by the Nanobiotechnology Laboratory, Federal University of Uberlândia, Brazil.

**Table 2 vetsci-09-00158-t002:** Age and breed of dogs with pyometra that was positive for *E. coli*.

Animal	Breed	Age (Years)	Uterine Secretion	Urine
01	Mongrel	8	+	+
02	Mongrel	12	+	+
03	Mongrel	14	+	+
04	Pinscher	8	+	+
05	Rottweiler	10	+	+
06	BassetHound	10	+	+
07	Mongrel	1	+	−
08	Mongrel	1	+	−
09	Mongrel	4	+	−
10	Mongrel	11	+	−
11	Mongrel	11	+	−
12	Mongrel	13	+	−
13	Poodle	9	+	−
14	Poodle	10	+	−
15	Poodle	10	+	−
16	Poodle	11	+	−
17	Poodle	11	+	−
18	Pitt Bull	8	+	−
19	Pitt Bull	9	+	−
20	Pitt Bull	10	+	−
21	Rottweiler	9	+	−
22	Rottweiler	10	+	−
23	BassetHound	6	+	−
24	Sharpei	4	+	−
25	Akita	8	+	−
26	Mongrel	7	−	+

Results from samples of 52 animals. +: Positive samples for *E. coli*, −: negative samples for *E. coli*.

**Table 3 vetsci-09-00158-t003:** Prevalence of virulence genes and profile of virulence of *E. coli* from female dogs with pyometra.

Prevalence of Virulence Genes	Presence—N (%)
*Pap*	15 (46.9)
*Hly*	17 (53.1)
*Afa*	19 (59.4)
*Cnf*	22 (68.8)
*Sfa*	31 (96.9)
**Virulence Profiles**	**Presence—N (%)**
V1: *sfa*	8 (25.0)
V2: *pap*, *cnf*	1 (3.1)
V3: *afa*, *sfa*	1 (3.1)
V4: *cnf*, *sfa*	1 (3.1)
V5: *pap*, *afa*, *sfa*	1 (3.1)
V6: *cnf*, *hly*, *sfa*	1 (3.1)
V7: *cnf*, *afa*, *sfa*	3 (9.4)
V8: *pap*, *cnf*, *hly*, *sfa*	2 (6.2)
V9: *cnf*, *afa*, *hly*, *sfa*	3 (9.4)
V10: all genes	11 (34.4)

N (%) frequency and percentage of positivity for each gene or virulence profile.

## Data Availability

All data are included in the article.

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
