# Peer review of "Phylogeny and Virulence Factors of Escherichia coli Isolated from Dogs with Pyometra"

_vetsci, 2022, doi:10.3390/vetsci9040158_

Round 1

Reviewer 1 Report

I believe the manuscript is significantly improved with the exclusion of the antimicrobial susceptibility results. This is a much more clear and well-defined story.  The authors should be commended for their hard work and I believe that this paper is now ready for publication.

Author Response

We appreciate the feedback, and also the availability to evaluate the article.

Reviewer 2 Report

The manuscript regarding Phylogeny and virulence factors of Escherichia coli isolated from dogs with pyometra  has been significantly improved and
shoould be publish in vetsci.

Author Response

We appreciate the feedback, and also the availability to evaluate the article.

This manuscript is a resubmission of an earlier submission. The following is a list of the peer review reports and author responses from that submission.

Round 1

Reviewer 1 Report

The study conducted is very interesting.

The results are in line with the objective of the study.

The number of subjects taken into consideration for the study is a reasonable number, this number is suitable to support a valid statistical analysis.Escherichia coli is the primary causative agent of pyometra in dogs, very interesting is the connection between uterine infections and urinary tract infections, even more interesting to know which causes the other in the future, as the pathologies of the urinary tract are very widespread.the most important aspect of the work is that of antimicrobial resistance, a topic of global importance that afflicts medicine the use of the antibiogram as a test for successful therapeutic results and to counter the indiscriminate use of antibiotics in pets is now essential in veterinary medicine.

Reviewer 2 Report

The manuscript was very well written and provides information regardings a theme that is very common in veterinary clinics but still very scarce information available.

Overalll is well presented, organized and writeen.

Some minor suggestions:

  1. reference section please correct the different types of letters.
  2. The text should be read again and correct some minor grammatical mistakes;
  3. table 3 is supposed to appear with the background grey?
  4. material and methods: what was the period in which the samples were collected

Reviewer 3 Report

This manuscript attempted to characterize isolates of E. coli from dogs diagnosed with Pyometra. Isolates were characterized by virulence gene PCR, AST and RAPD-PCR.  While there has clearly been a lot of work completed, unfortunately the manuscript, as written, is not acceptable for publication.  I have several major concerns that must be addressed before the publication can be accepted which are outlined below:

  1. The background is insufficient and does not explain why this study was performed and what this study would add to the veterinary literature. Currently, I have concerns that this study does not add much to the literature. Please highlight what is novel or unique.
  2. The methods are lacking some key elements which would prevent replication of the study.
  3. For animals with bacteriuria, clinical signs have not been evaluated in order to deem the diagnosis a UTI or simply bacteriuria (ACVIM consensus statement- Weese et al 2019).
  4. The antimicrobial resistance testing does not follow veterinary standards and lacks clinical relevance in some cases. In the reviewers opinion, I would not include these data unless they can be performed by broth microdilution or E-test for clinically relevant drugs. Currently, it weakens the paper. I would exclude these data and focus on the virulence data which is far more compelling.
  5. The RAPD-PCR study is by far the best part of the manuscript but lacks integration with the additional data. I would recommend overlaying resistance genes and determining if those are associated with particular clades/clusters other then the isolates from the same animals.
  6. References lack corresponding numbers- I have not been able to assess the appropriateness for many statements. It is possible I would have concerns over some statements but I am not able to “fact check.” This was particularly challenging throughout the discussion and I do not feel I am able to properly review.
  7. The manuscript, unfortunately, suffers throughout from a lack of proper English and verbose or awkward syntax. It is very difficult to read, especially throughout the discussion. If possible, I would recommend revision by a person or service that can improve this.

A line-by-line evaluation can be found below.

Line 34- Syntax is confusing; recommend rewriting sentence.

Line 36- This statement is probably only true regionally based on spay rates- please elaborate or be more specific about the epidemiology.

Line 39- Please expand on this statement. Does this mean that the same types cause UTI and pyometra or it’s the same strain in dogs with concurrent pyometra and UTI?

Line 42- This paragraph also needs some expansion. Expand on pathogenesis (if its been demonstrated) or describe previously demonstrated virulence factors and whether these have been evaluated in pyometra isolates specifically.

Line 46- Syntax is confusing; recommend rewriting sentence.

Line 51- I think this is too broad of a statement to include without specific evidence of this occurring in pyometra.

Line 57- Move to results

Line 58- Were patients at this hospital?  Or were owned by the university?

Line 62- Change to “A diagnosis of pyometra was made based on …”

Line 66- Was it attempted for all animals?  Were animals examined for clinical signs of UTI?

Line 71- BHI Agar or broth?  Was enrichment performed? Please also include incubation conditions (temp and atmospheric conditions and time).

Line 72- Rephrase; suggested “a commercial kit was used for identification of bacteria by biochemical profiling (LIST)”

Line 76- This is an inappropriate reference; there are veterinary specific documents that should be used and not the human m100 (Vet01 and Vet 01s). A lot of drugs on here are not found in the Vet01s and many do not have zone diameters for clinically relevant. Insufficient details provided about media. “Sulfonamide” is not a drug (it’s a class)- which drug was tested? This is particularly important for drugs like ampicillin for which KB disc diffusion can not be performed for soft tissue infections like pyometra. I recommend a thorough review of these requirements if this data is to be included. Several of these drugs lack clinical relevance in veterinary medicine or for systemic drug use) and their inclusion is confusing (i.e. nalidixic acid, ciprofloxacin, neomycin). Testing of Gram negative bacteria against erythromycin is not clinically relevant.

Line 83- There is not enough detail here for extraction or thermocycling; gel electrophoresis.  References missing from table. Also, it is unclear why these genes were selected and not others. Did not describe positive or negative control strategies.

Line 89- “assessed” instead of “performed”; where are details on thermocycling and electrophoresis.

Line 99- “submitted” is a strange word to use.

Line 109- Positivity for E. coli should be out of total specimens tested.

Line 111- Syntax is confusing; recommend rewriting sentence.

Line 118- Very wordy and difficult to understand.

Line 121- Define MDR; Inclusion of clinically irrelevant drugs in the evaluation of these drugs as part of this definition inflates actual rates.

Line 131-142- Inclusion of the total number after each number is not needed and makes it very difficult to read. Suggest throughout: “The sfa gene was found in 31/32 E. coli isolates (96.9%) of which 24 were from uterine samples and 7 from urine” These data may be better just presented in a table and key results highlighted in text.

Line 133- Reference these other studies please.

Line 143- This section may no longer apply once proper testing guidelines are followed.

Line 168- I don’t understand this sentence

Line 170- Split out by uterine and urine.

Line 175-186- A large proportion of this discussion is likely irrelevant.

Round 2

Reviewer 3 Report

Thank you for your significant revisions.  I appreciate the time and effort done to improve this manuscript.  It is much easier to read given the significant proofreading and grammatical changes. I also am sympathetic to funding challenges and celebrate the work of these authors in a resource limited setting.

I still have a major issue with the use of the m100 standards alone for this study. 

You have used zone diameters used to detect CLINICAL resistance and therefore should be used for the species host. I understand the "comparative" nature you are going for but this is incorrectly applied. If you are looking for a mechanism of resistance use of an ECOFF would be the appropriate approach, but can not be done with Kirby-Bauer. This can and will be misleading to readership with regards to the interpretation of the results. For example, if applying the appropraite clinical breakpoints, no E. coli isolates from outside of the urinary tract from dogs are clinically susceptible to ampicillin, but your paper suggests otherwise.

I understand you have found citations where this is done; however, that does not make it correct or acceptable to this reviewer. Bad practices make it into the literature and simply citing them does not absolve you of correct methodology. I find it slightly frustrating that no attempts were made to adjust this after my initial review.

I do believe that there is one option that will fit the limitations you have described to me. In your manuscript please:

-Explicitly explain that you used the m100 (human) zone diameters for the purposes of detecting suspected resistant phenotypes that may be reflective presence of a resistance gene. 

-Explain that this differs from the veterinary clinical zone diameters that are used in a diagnostic lab setting and are the only appropriate way of determining therapeutic approach.

-Include a second set of interpretations based on animal zone diameters for drugs where this is possible. It may be most appropriate to provide this in the supplemental data. No additional costs will be incurred in this way. A free copy of the Vet01s can be found here: https://clsi.org/standards/products/free-resources/access-our-free-resources/ Simply click guest access in the top righthand corner.

Thank you again for your revisions and I look forward to a future version where I can provide final feedback.
